

# The combined effects of temperature and aromatase inhibitor on metamorphosis, growth, locomotion, and sex ratio of tiger frog (*Hoplobatrachus rugulosus*) tadpoles

Yun Tang[1,2], Zhi-Qiang Chen[1,3], You-Fu Lin[1,4], Jing-Yi Chen[1], Guo-Hua Ding[1] and Xiang Ji[2]

[1] Laboratory of Amphibian Diversity Investigation, College of Ecology, Lishui University, Lishui, Zhejiang, P.R. China
[2] College of Life Sciences, Nanjing Normal University, Nanjing, Jiangsu, P.R. China
[3] College of Animal Science and Technology, Zhejiang A & F University, Lin'an, Zhejiang, P.R. China
[4] College of Forestry, Nanjing Forestry University, Nanjing, Jiangsu, P.R. China

## ABSTRACT

**Background:** The tiger frog (*Hoplobatrachus rugulosus*) is widely raised by many farms in southern region of China as an economically edible frog. The growth, development, and sexual differentiation of amphibians are influenced by temperature and steroid hormone level. However, the problem of hormone residues is caused by the addition of exogenous hormones in frog breeding, it is worth considering whether non-sterol aromatase inhibitors can be used instead of hormones.

**Methods:** In our study, *H. rugulosus* tadpoles were subjected to two water temperatures (29 °C and 34 °C) and three letrozole concentrations in the feed (0, 0.1 and 1 mg/g) to examine the effects of temperature, aromatase inhibitor and their interaction on metamorphosis, locomotion, and sex ratios. A *G*-test and contingency table were used to analyze the metamorphosis rate of tadpoles and the survival rate of froglets after feeding for 90 days. A *G*-test was also used to analyze sex ratios in different treatment groups.

**Results:** Metamorphosis time and body size (snout–vent length, body mass and condition factor) were significantly different between the two temperature treatments. Metamorphosis time was longer and body size was increased at 29 °C compared to those at 34 °C. Letrozole concentration and the temperature × letrozole interaction did not affect these variables. The jumping distance of froglets following metamorphosis was positively associated with the condition factor; when controlling for condition factor, jumping distance was not affected by temperature, letrozole concentration and their interaction. Temperature and letrozole concentration also did not affect metamorphosis and survival rate. Sex ratio of the control group (0 mg/g letrozole) was 1:1 at 29 °C, but there were more males at 34 °C. The sex ratios of *H. rugulosus* treated with letrozole at 29 °C and 34 °C were significantly biased toward males, and male ratio increased as letrozole concentration increased. Furthermore, more males were produced at 34 °C than at 29 °C at each letrozole concentration.

Corresponding author
Guo-Hua Ding, guwoding@qq.com, guwoding@lsu.edu.cn

## INTRODUCTION

The growth and sex differentiation of amphibians are often influenced by the environment, and the effect of temperature has received much attention from researchers. Previous studies have found that the hatching success rate and survival rate of amphibians are significantly affected by temperature (*Wang & Li, 2007*; *Fu & Xu, 2014*). During the development of tadpoles, high temperature accelerates the growth rate and reduces the duration of metamorphosis and time to sexual maturation (*Wang & Li, 2007*; *Liu et al., 2006*; *Wang, Li & Zhang, 2005*). However, high temperatures can lead to malformations or even death, while low temperatures can lead to the failure of metamorphosis (*Wang & Li, 2007*; *Wang, Li & Zhang, 2005*). The growth and development of amphibians is also reflected by locomotion, previous studies have focused on the relationship between temperature and locomotion (*Huey & Stevenson, 1979*; *Tracy, 1979*; *Rome, Stevens & John-Alder, 1992*). In addition, gonadal differentiation in amphibians is not completely controlled by genes, and environmental factors such as temperature affect gonadal differentiation to determine phenotypic sex (*Tompsett et al., 2013*). Previous studies reported that tadpoles experiencing extreme temperatures exhibited a significant shift in phenotypic sex ratio of the offspring (*Nakamura, 2009*). Tadpoles from families such as Bufonidae, Ranidae, and Dicroglossidae are biased toward developing as males at high temperatures and females at low temperatures (*Li et al., 2001*; *Li, You & Lin, 2007*; *Dournon, Houillon & Pieau, 1990*; *Piquet, 1930*; *Yoshikura, 1959*; *Hsü, Yü & Liang, 1971*; *Fu, 2010*). However, the sensitivity of sex ratio variation to temperature is not consistent in different species. Moreover, gonadal differentiation is more significantly affected by temperature when tadpoles develop to a certain period, and the period is called thermosensitive period (*Kraak & Pen, 2002*).

In addition to temperature, previous studies have shown that steroid hormones can affect the metamorphosis of amphibians (*Hayes, Chan & Licht, 1993*; *Hayes, 1997*). However, few studies have assessed the effect of steroid hormones on amphibian growth, development, and locomotion with most studies focusing on effects on gonad development and phenotypic sex (*Li & Lin, 2000*; *Nakamura, 2009*, *2010*, *2013*). Generally, exogenous testosterone or dihydrotestosterone lead to masculinization of females (*Nishioka, Miura & Saitoh, 1993*; *Martyniuk, Bissegger & Langlois, 2013*), while exogenous estradiol can feminize males into females (*Zhang & Witschi, 1956*), and even offset the temperature-induced sex reversal effect. For example, adding estradiol to water at high temperature does not skew the sex ratio in amphibian populations (*Nakamura, 2009*). However, the effects are not consistent on different species and may even be variable within a species in a dose-dependent manner (*Nakamura, 2009*; *Piprek et al., 2012*; *Stephanie et al., 2016*). Researchers have found that during steroid hormone synthesis in vertebrates, Cytochrome P450 17α-hydroxylase and 17,20 lyase (CYP17) can promote the conversion of progesterone to dehydroepiandrosterone in amphibians

(*Maruo et al., 2008*), unregulated gene expression in indifferent gonads of males, and then maintain this at a high level (*Iwade et al., 2008*); cytochrome P450 aromatase (CYP19) can transform testosterone into estradiol (*Maruo et al., 2008*) and is expressed at a higher level in the undifferentiated gonads of females (*Kuntz et al., 2003a*, *2003b*; *Kato et al., 2004*). For example, in female tadpoles injected with testosterone, the activity of CYP17 is enhanced and that of CYP19 is inhibited to a certain extent under conditions of high estradiol concentration (*Yoshikura, 1959*).

Most previous studies have used exogenous testosterone and estradiol to explore their influence on sex differentiation (*Hayes, Chan & Licht, 1993*; *Hayes, 1997*; *Oike et al., 2016*). In fact, the levels of testosterone and estradiol can be directly regulated by altering the activity of aromatase in the steroid hormone synthesis pathway in animals (*Foidart et al., 1994*; *Nathan et al., 2001*; *Urbatzka, Lutz & Kloas, 2007*). Aromatase inhibitors comprise a class of synthetic drugs that, apart from inhibit the activity of aromatase (*Li et al., 2007*), block the transformation of testosterone to estradiol, and reverse the sex transition from female to male or masculinize the gonads (*Yu et al., 1993*; *Chardard & Dournon, 1999*; *Miyata & Kubo, 2000*). Previously, the effects of aromatase inhibitors on steroid hormone levels and gonadal development has been increasingly reported in Ribbed Newt *Pleurodeles waltl* (*Chardard & Dournon, 1999*) and American Bullfrog *Rana catesbeiana* (*Yu et al., 1993*). Studies have shown that several aromatase inhibitors (e.g., fadrozole and 4-hydroxyandrostenedion) can induce the masculinization of amphibian ovaries (*Chardard & Dournon, 1999*; *Duarte-Guterman et al., 2009*; *Miyata & Kubo, 2000*; *Olmstead et al., 2009*; *Yu et al., 1993*), resulting in intersexed gonads or even complete masculinization (*Chardard & Dournon, 1999*; *Olmstead et al., 2009*). In contrast, other aromatase inhibitors (e.g., aminoglutethimide) have no effect on amphibian gonads (*Chardard & Dournon, 1999*), while miconazole has been found to have a toxic effect on amphibian tadpoles (*Chardard & Dournon, 1999*). In addition, a close correlation between testosterone levels and muscle strength was reported in humans (*Nam et al., 2018*), which suggests that testosterone might affect the locomotion of animals by improving muscle strength. Aromatase inhibitors can regulate testosterone levels in organisms, but whether aromatase inhibitors can affect the locomotion of animals needs to be tested.

As stated, numerous studies have reported that temperature, steroid hormones, and aromatase inhibitors play important roles in amphibian growth or sex development (*Hayes, Chan & Licht, 1993*; *Hayes, 1997*; *Chardard & Dournon, 1999*), but these factors might interact during amphibian life, and such interactions still need to be studied. Early studies have reported interactions between temperature and steroids, with resulting effects on amphibian larval growth, development, and metamorphosis (*Hayes, Chan & Licht, 1993*); however, the effects of these interactions on amphibian sex development have rarely been assessed. Aromatase inhibitors do not exist in nature, but they have become more widely used in recent years because they can affect the levels of endogenous steroid hormones and are associated with better hormonal regulation than exogenous steroid hormones (*Miyata & Kubo, 2000*; *Olmstead et al., 2009*; *Shen et al., 2013*; *Singh et al., 2015*). Given the state of research on aromatase inhibitors and the potential effects of steroids on anuran larval growth and development, an investigation of the interactive

effects of temperature and aromatase inhibitors on growth and sex development is warranted. Fadrozole is an aromatase commonly used for amphibians (*Olmstead et al., 2009*), but in other animals like fish and reptiles (*Noëlle et al., 1995*; *Shen et al., 2013*; *Singh et al., 2015*), the aromatase inhibitor letrozole (*Lamb & Adkins, 1998*) prevents the conversion of testosterone to estradiol, thereby altering the levels of steroid hormones in organisms. Letrozole has shown high selectivity for and the potential to inhibit aromatase (*Shen et al., 2013*). Moreover, it was found to exert a stronger effect than fadrozole in the European Pond Turtle *Emys orbicularis* (*Noëlle et al., 1995*), but it has rarely been used as an aromatase inhibitor in amphibians.

*Hoplobatrachus rugulosus*, a large robust dicroglossid frog, is listed in Appendix II of CITES as a national Class II protected species in China (*Fei, Ye & Jiang, 2012*). It is widespread form the southern region of the Yangtze River within China to Myanmar, Laos, Vietnam, Cambodia and Thailand, and inhabits a variety of lowland habitats including intermittent freshwater marshes and seasonally flooded agricultural land (*Fei, Ye & Jiang, 2012*). *Hoplobatrachus rugulosus* is considered an economically edible frog species in China, owing to its delicious and nutritious meat (*Ding et al., 2015*). In China, there are many frog farms that raise *H. rugulosus* since 1980s (*Zhan & Yang, 2012*). These farms should consider the production efficiency and economic efficiency with different sexes of frogs, and it is known that sex ratio bias induced by temperature has a high practical value, but the economic efficiency is not as good as that induced by hormones (*Fu, 2010*). However, hormone residues are harmful, and it is worth considering whether non-sterol aromatase inhibitors can be used instead of hormones. In our study, the effects of different temperatures and letrozole concentrations on the metamorphosis, growth, locomotion, and sex of *H. rugulosus* tadpoles were studied. Furthermore, the combined effects of environmental temperature and aromatase inhibitors on the phenotypic traits of *H. rugulosus* tadpoles were also evaluated. The purpose of our study was to elucidate the internal and external factors influencing the growth, development, and sexual differentiation of *H. rugulosus*, and to provide a basic reference for the artificial breeding of this species.

## MATERIALS AND METHODS

### Animal collection and treatment

Our experimental procedures were specifically approved by the Animal Research Ethics Committee of College of Ecology in Lishui University (Permit No. AREC-CELSU 201505-001).

In June 2015, four clutches of fertilized eggs of *H. rugulosus* were collected from the amphibian laboratory of Lishui University. They were placed in plastic bins (length × width × height = 50 cm × 40 cm × 35 cm) with 30 L water, and the boxes were moved to an outdoor shelter. Through natural incubation, the fertilized eggs developed into tadpoles at Gosner 25. Then, 135 tadpoles from each clutch were randomly selected and mixed. All 540 tadpoles were divided into six groups and placed into six food-grade polypropylene plastic bins with 50 L of aerated water. The population density of *H. rugulosus* tadpoles will

significantly affect their metamorphosis (*Ding et al., 2015*); therefore, the initial density was maintained at 1.8 individuals/L.

Previous studies on the effects of aromatase inhibitors in amphibian species were conducted by mixing the aromatase inhibitors into feed (*Chardard & Dournon, 1999*), putting the aromatase inhibitors in the water (*Duarte-Guterman et al., 2009*), or implanting the capsules with aromatase inhibitors on the mesenteries of tadpoles (*Yu et al., 1993*). Letrozole is insoluble in water, and it is difficult to implant the capsules on the mesenteries of tadpoles. Therefore, in our study, we decided to mix the letrozole into the feed. Before the experiment, 0.02 g and 0.2 g letrozole was dissolved in 100 mL of anhydrous ethanol, and the two treatment solutions were evenly sprayed and stirred into 200 g frog feed (Ningbo Tech-Bank Co., Ltd., Ningbo, China; water ≤12.0, crude protein ≥42.0, crude fat ≥3.0, crude fiber ≤4.0, crude ash ≤18.0, calcium ≥1.5, total phosphorus ≥1.0, and salt ≤3.0). The feed for the control group was only sprayed with 100 mL anhydrous ethanol. The three kinds of feeds were then oven heated at 50 °C for 2 h to completely volatilize the ethanol, and the feeds with letrozole concentration of 0 mg/g, 0.1 mg/g and 1 mg/g were prepared for later use. In previous studies, researchers have found that the body temperature preference for the growth and development of *H. rugulosus* tadpoles is 28.2 °C (*Fan, Lei & Lin, 2012*). Another study found that the sex ratio was biased toward males at 30 °C and that 100% masculinization occurred at 35 °C (*Fu, 2010*), suggesting that high temperatures can make *H. rugulosus* tadpoles produce more male offspring. Therefore, we used 29 °C and 34 °C for tadpole feeding experiments based on these previous studies. There were two (water temperature: 29 °C and 34 °C) × three (letrozole concentration in feed: 0 mg/g, 0.1 mg/g and 1 mg/g) experimental treatments designed. Six bins were used, and the water temperature inside the bins was controlled by two 300 W heating rods. Three bins of tadpoles at each temperature were fed with different letrozole concentration feeds at 8:00 daily. During the first week of the experiment, 0.3 g feed was added to each bin daily. After the first week, 10 tadpoles were randomly selected from each bin and removed with a net every 2 days. These were weighed after towel drying, and 10% of the mean weight of the tadpole was used as the feed mass for the next 3 days. The water and excreta at the bottom of the bins was pumped out every 2 days and replaced with the same amount of fresh aerated water. The water volume was determined by the number of surviving tadpoles, so that the tadpole density was maintained at 1.8 individuals/L. The amount of water changed each time was about half of the whole bin.

## Data measurement

After complete metamorphosis of tadpoles (Gosner 46) (*Gosner, 1960*), metamorphosis time of each individual and metamorphosis number were recorded, and the snout–vent length (SVL, the distance from the snout to the cloaca orifice) and body mass of the first 20 froglets to complete metamorphosis in each treatment group were measured with a digital caliper and electronic scale. Only comparing the SVL or weight was not enough to reflect the overall body size of *H. rugulosus* and the condition factor was defined as the body mass divided by the SVL (*Hu et al., 2019*). Therefore, we used the condition factor as

the overall indicator of body size. Then, the froglets were put into a lidded plastic bowl (diameter, 10 cm) with a saturated sponge and stood for 1 h at 25–28 °C. After that, the feet of the froglets were colored with green pigment and placed on flat ground without obstacles. Then, the froglets were touched on the tail bone with a glass rod to initiate jumping onto a white gauze three times in a row (jumping from where they landed from the previous jump), and the distance was measured with a digital caliper (± 0.01 mm). The average distance was taken as the jumping ability. PIT animal tags (HT100, 0.02 g, length × diameter = 7.5 mm × 1.2 mm, Guangzhou Hongteng Barcode Technology Co. Ltd., Guangzhou, China) were subcutaneously injected to mark individual froglets. After injection, a sponge saturated with water was placed in the cage, and the froglet was placed in the cage to recover. The froglets were returned to the pool to continue feeding after the wound healed. The froglets were reared in separate outdoor breeding ponds (length × width × height = 3 m × 1.8 m × 1 m) according to the different treatments, and the outdoor environment was simulated in the ponds (5 cm silt on the bottom; 10 cm water depth). *Myriophyllum verticillatum* and *Hydrocotyle vulgaris* were planted in the ponds, and *Azolla imbricata* floated on the water surface. To determine the feed mass, 10% of the mean weight of the froglets × the froglet number was calculated every 3 days, and remaining feed was removed after 3 h. The number of surviving individuals was recorded after 90 days of feeding and used to calculate the survival rate for the froglets. Some individuals died after metamorphosis, and we randomly selected some of them for gonadal dissection (5–10 dead froglets in each treatment) to estimate the number of male and female individuals in each treatment surviving after 90 days. Males were considered to be those with a pair of vocal sacs, and the others were considered females. If the body length of an individual without vocal sacs was <55 cm, then the sex was determined by anatomical observation of the gonads after euthanasia with MS-222 (400 ppm). The male ratio of each treatment group was calculated by combining the estimated number of male and female individuals who died and the number of male and female individuals who survived after 90 days.

## Statistical analysis

Before further statistical analysis, normality and homogeneity of all data were verified by the Kolmogorov–Smirnov test and the Bartlett's test, respectively. A log likelihood-ratio test (*G*-test) and contingency table were used to evaluate the metamorphosis rate and survival rate of froglets after feeding for 90 days. The *G*-test was used to analyze the sex ratios of *H. rugulosus* in different treatment groups. Linear regression analysis was used to analyze the relationship between jumping distance and condition factor. With temperature and aromatase inhibitor concentration as factors, two-way ANOVA was used to analyze the differences in metamorphosis time, individual size, and residual value of jumping distance against condition factor among different treatments. Tukey multiple comparisons were used to analyze the differences. All statistical tests were performed using the STATISTICA software package (version 6.0). All results are presented as mean ± SE, and the differences were considered statistically significant at $P < 0.05$.

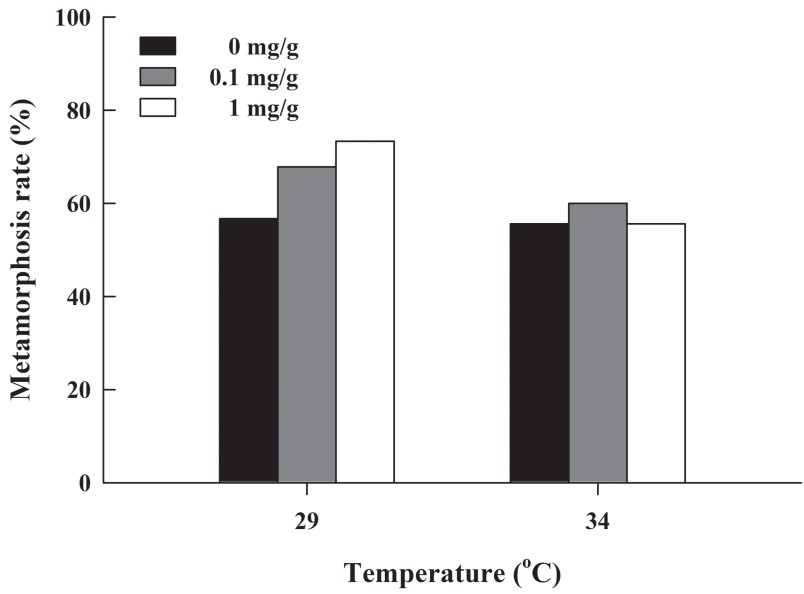

**Figure 1 Metamorphosis rate of *H. rugulosus* tadpoles from treatments involving 2 temperatures × 3 letrozole concentrations.**

## RESULTS

The metamorphosis rate of *H. rugulosus* tadpoles under different treatments ranged from 55.6% to 3.3% (61.5 ± 3.0% average). The set temperature and letrozole concentration did not affect the metamorphosis rate of tadpoles ($G = 10.74$, df = 5, $P > 0.05$; Fig. 1). The metamorphosis time, SVL, body mass, and condition factor after complete metamorphosis were significantly different between the two temperatures. Treatment at 29 °C prolonged the metamorphosis time and increased the SVL, body mass, and condition factor of the froglets compared with those at 34 °C. However, different letrozole concentration and the interaction between temperature and letrozole concentration did not affect the four indicators (Table 1). The jumping distance of froglets was positively correlated with condition factor ($F_{1, 118} = 13.88$, $P < 0.001$; Fig. 2A). After controlling for the effect of the condition factor, jumping distance was not affected by temperature ($F_{1, 114} = 0.92$, $P = 0.339$), letrozole concentration ($F_{2, 114} = 2.04$, $P = 0.134$), or their interaction ($F_{2, 114} = 2.96$, $P = 0.056$) (Fig. 2B).

There was no significant difference in the froglets survival rate of *H. rugulosus* in the six treatment groups after 90 days of feeding ($G = 2.83$, df = 5, $P = 0.727$), with an average survival rate of 49.7 ± 2.1% (42–56.1%; Fig. 3A). Under the non-letrozole treatment, the sex ratio of *H. rugulosus* froglets was maintained at 1:1 at 29 °C (54.9% male, 45.1% female; $G = 0.49$, df = 1, $P = 0.483$). However, the proportion of males was higher at 34 °C (86%; $G = 28.82$, df = 1, $P < 0.001$) (Fig. 3B). Exposed to letrozole, the sex ratio of froglets at both 29 °C and 34 °C was significantly biased toward males (0.1 mg/g at 29 °C: 83.6%, 0.1 mg/g at 34 °C: 98.1%, 1 mg/g at 29 °C: 92.4%, 1 mg/g at 34 °C: 100%; all $P < 0.001$) (Fig. 3B). The male ratio increased with letrozole concentration (both $P < 0.01$) at both temperatures, while more males were produced at 34 °C than at 29 °C at each letrozole concentration (both $P < 0.05$) (Fig. 3B).

**Table 1 Descriptive statistics, expressed as means ± SE (range), for metamorphosis time, snout–vent length, body mass and condition factor of froglets, and results of two-way ANOVAs.**

| Temperature (°C) | Letrozole concentration (mg/g) | Metamorphosis time (days) | Snout–vent length (mm) | Body mass (g) | Condition factor (g/mm) |
|---|---|---|---|---|---|
| 29 | 0 | 26.1 ± 0.3 (23–31) | 22.0 ± 0.3 (19.9–24.1) | 1.48 ± 0.05 (1.20–2.02) | 0.067 ± 0.002 (0.055–0.085) |
| | 0.1 | 26.3 ± 0.3 (23–35) | 22.1 ± 0.4 (18.6–25.0) | 1.56 ± 0.07 (1.00–2.34) | 0.070 ± 0.002 (0.054–0.093) |
| | 1 | 26.0 ± 0.3 (23–30) | 22.0 ± 0.3 (20.2–24.8) | 1.50 ± 0.09 (1.03–2.34) | 0.068 ± 0.003 (0.051–0.096) |
| 34 | 0 | 20.8 ± 0.3 (17–25) | 21.2 ± 0.2 (19.6–23.2) | 1.36 ± 0.03 (1.11–1.69) | 0.064 ± 0.002 (0.051–0.082) |
| | 0.1 | 21.0 ± 0.2 (17–28) | 21.5 ± 0.3 (19.1–24.7) | 1.32 ± 0.03 (1.09–1.67) | 0.062 ± 0.001 (0.051–0.076) |
| | 1 | 21.3 ± 0.3 (17–28) | 21.3 ± 0.3 (18.6–23.1) | 1.39 ± 0.04 (1.11–1.72) | 0.065 ± 0.002 (0.052–0.081) |
| Statistical results | Temperature | $F_{1,\ 326} = 463.79$ $P < 0.001$; T29 > T34 | $F_{1,\ 114} = 7.98$ $P < 0.01$; T29 > T34 | $F_{1,\ 114} = 12.35$ $P < 0.001$; T29 > T34 | $F_{1,\ 114} = 7.34$ $P < 0.01$; T29 > T34 |
| | Letrozole concentration | $F_{2,\ 326} = 0.29$ $P = 0.750$ | $F_{2,\ 114} = 0.23$ $P = 0.794$ | $F_{2,\ 114} = 0.11$ $P = 0.896$ | $F_{2,\ 114} = 0.11$ $P = 0.893$ |
| | Interaction | $F_{2,\ 326} = 0.82$ $P = 0.442$ | $F_{2,\ 114} = 0.03$ $P = 0.966$ | $F_{2,\ 114} = 0.91$ $P = 0.406$ | $F_{2,\ 114} = 1.53$ $P = 0.221$ |

**Note:**
Tukey's post hoc comparison was performed on the trait that differed between the two temperature treatments. T29: 29 °C, T34: 34 °C.

## DISCUSSION

The influence of temperature on the life history of ectotherms has been previously studied by several researchers (*Roff, 1990*; *Stearns, 1992*; *Charnov, 2004*; *Nie et al., 2007*), and it has been reported on poikilothermic species such as Eurasian Perch *Perca fluviatilis* (*Sandstrom, 1995*), Japanese Medaka *Oryzias latipes* (*Hemmer-Brepson et al., 2004*) and Multiocellated Racerunner *Eremias multiocellata* (*Li et al., 2011*). Metamorphosis is an important developmental stage in amphibians (*Meng, 2019*). Here, we focused on the effects of temperature, aromatase inhibitor and their interaction on the metamorphosis of *H. rugulosus*.

Our results showed that the metamorphosis time of *H. rugulosus* tadpoles at high temperature was shorter than that at low temperature, but the body size of froglets decreased. These results are similar to those from previous studies (*Álvarez & Nicieza, 2002*; *Liu et al., 2006*; *Gomez-Mestre & Buchholz, 2006*), suggesting that temperature is closely related to the growth of amphibians; specifically, higher temperatures might increase the metabolic activity of tadpoles and accelerate their development. However, growth is affected owing to the shorter development time (*Wang & Li, 2007*; *Wang & Wang, 2008*), and this shorter time leads to less energy being accumulated and, consequently, smaller froglets. In addition to temperature, our results also showed that treatment with letrozole at different concentrations had no significant effect on metamorphosis time or body size of *H. rugulosus* froglets, which suggests that letrozole concentration does not significantly affect their growth or development. Furthermore, the results showed that the metamorphosis rate of tadpoles and the survival rate of froglets were not significantly affected by different temperatures or letrozole concentrations. However, previous studies reported that the metamorphosis rate of Chinese Brown Frog *Rana chensinensis* and Asiatic Toad *Bufo gargarizans* increased with increasing

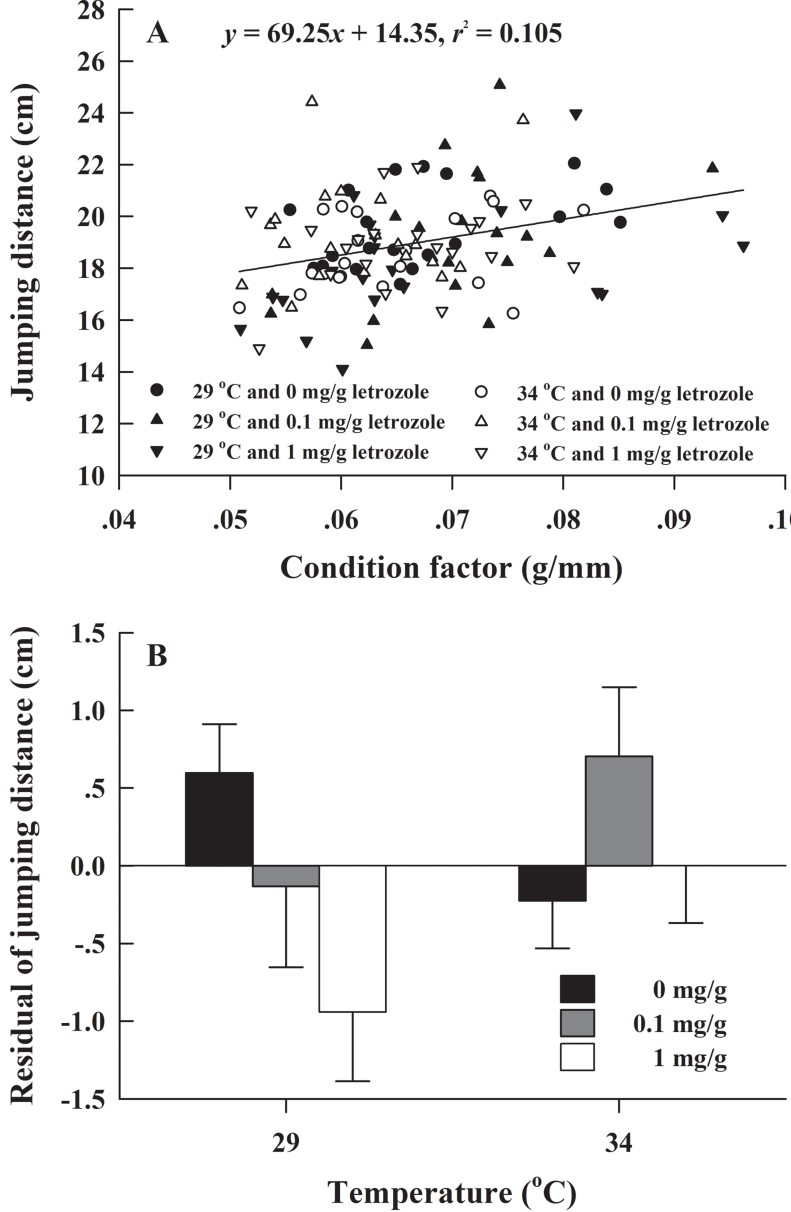

**Figure 2 (A) Correlation of jumping distance with condition factor and (B) mean values (+SE) for residual of jumping distance of *H. rugulosus* froglets at complete metamorphosis from treatments involving 2 temperatures × 3 letrozole concentrations.** Regression equation and coefficient are indicated in the figure.                

temperature (*Wang, Li & Zhang, 2005*), which is inconsistent with our results, suggesting that temperature is independent of the metamorphosis rate of *H. rugulosus*. A possible reason for this discrepancy is that the temperature range used in the present study might not have been broad enough to detect an effect as it was only 5 °C (29–34 °C), whereas that in *Wang, Li & Zhang (2005)* was 20 °C (5 °C, 15 °C and 25 °C). Thus, more data is needed to determine whether temperature is related to the metamorphosis rate of amphibians. Previous studies on the effects of aromatase inhibitors on amphibians mainly

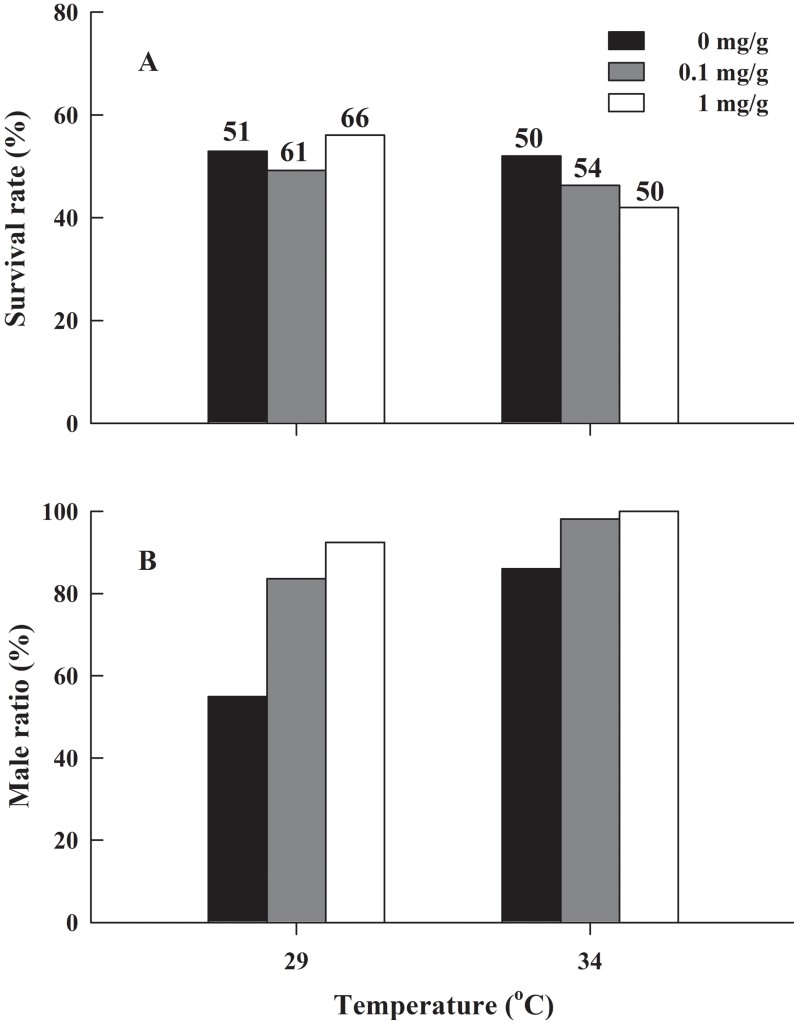

**Figure 3** (A) Survival rate and (B) male ratio at 90 days after complete metamorphosis in *H. rugulosus* from treatments involving 2 temperatures × 3 letrozole concentrations. The sample sizes are indicated in the figure.

focused on their sexual development. Further experiments on other aromatase inhibitors are needed to explore the effects of aromatase inhibitors on the metamorphosis development of amphibians. Although no replicate groups were included in our study, each treatment group included a mixture of tadpoles randomly selected from four different sources, thus increasing the validity and reliability of our results.

After controlling for the effect of the condition factor, temperature did not affect the jumping ability of *H. rugulosu*s. Previous studies on Green Frog *Rana clamitans* and Northern Leopard Frog *Rana pipiens* reported that their jump performance was relatively independent of temperature within a certain range (*Huey & Stevenson, 1979*; *Tracy, 1979*) suggesting that temperature within a specific range does not significantly affect the locomotion of *H. rugulosus*. In other amphibians (e.g., African Clawed Frog *Xenopus laevis*, Mudpuppy *Necturus maculosus, R. pipiens*, Spotted Grass Frog *Limnodynastes tasmaniensis* and Striped Marsh Frog *L. peronii*), the locomotion performance declined

rapidly at a very low or high temperature (*Putnam & Bennett, 1981*; *Miller, 1982*; *Hirano & Rome, 1984*; *Londos & Brook, 1988*; *Whitehead et al., 1989*; *Wilson, 2001*; *Gomes, Bevier & Navas, 2002*). However, in our study, the temperature was maintained constant, with no significant fluctuation, and the results indicated that the jumping ability of *H. rugulosus* is independent of temperature within the range set by us. Further studies are needed to explore the effect of different temperature treatments on amphibian locomotion ability. Similarly, letrozole concentration did not affect the jumping ability of *H. rugulosus*, but this evidence is not sufficient to conclude that aromatase inhibitors do not affect amphibian locomotion as there is research on other aromatase inhibitors. Therefore, further investigations are required to ascertain whether aromatase inhibitors influence amphibian locomotion.

The results regarding the sex ratio of *H. rugulosus* froglets suggested that the proportion of males reaches >80% at 34 °C. However, sex ratio was not evidently biased at 29 °C in the control group, which suggested that the gonads of *H. rugulosus* tadpoles are biased toward males at high temperature. These results are similar to those reported by *Fu (2010)*, and this phenomenon was observed in *R. chensinensis* (*Li et al., 2001*), Hong Kong Rice-paddy Frog *Fejervarya multistriata* (*Li, You & Lin, 2007*), and Giant Spiny Frog *Quasipaa spinosa* (*Mei et al., 2018*), suggesting that high temperature can cause male bias in most amphibians. The results of the present study also indicated that the sex ratio is biased toward males after letrozole treatment, and these results are similar to those based on Indian Skipper Frog *Euphlyctis cyanophlyctis* with the aromatase inhibitor formestane (*Phuge, 2018*). In a previous study, researchers implanted capsules in individuals to investigate the effects of aromatase inhibitors on sex hormones, and they also found that aromatase inhibitors at a certain concentration could inhibit the activity of ovarian aromatase, leading to the accumulation of testosterone and inducing the transformation of ovaries to testes (*Yu et al., 1993*). These results indicate that aromatase inhibitors can lead to the male bias. Previously, steroid hormones such as testosterone and estradiol were confirmed to change the sex ratio of amphibian offspring (*Nakamura, 2009*, *2010*, *2013*), but these trials used exogenous steroid hormones. In contrast, aromatase inhibitors can inhibit the transformation of testosterone to estradiol thus increasing endogenous testosterone levels, which could better reflect the regulatory mechanism of steroid hormones in vivo. In the present study, the proportion of males increased with increasing letrozole concentrations. In addition, at 29 °C, the proportion of males in the control group was 28.7% higher than that in the 0.1 mg/g letrozole treatment group. However, at 34 °C, the proportion of males in the control group was 12.1% higher than that in the 0.1 mg/g letrozole treatment group. Therefore, we speculate that temperature and letrozole interact to influence the sex ratio and that the effects of letrozole on the sex ratio are more obvious at lower temperatures.

## CONCLUSIONS

Our results showed that (1) high temperature can accelerate the growth and development of *H. rugulosus* tadpoles, shorten the metamorphosis time and increase the proportion of males; (2) although the tadpoles at low temperature grew slowly, the froglets after metamorphosis were larger; (3) letrozole can induce a male bias in the tadpoles of

*H. rugulosus*, and this male biased effect is more obvious at low temperature. While our results demonstrate the effects of temperature, letrozole concentration and their interaction on the growth, development and sex differentiation of tadpoles, the molecular mechanism should be further explored in future research.

## ACKNOWLEDGEMENTS

We would like to thank Ying-Ying Wang, Jing-Hao Zhu for their help during the research, and would like to thank Editage for English language editing.

### Funding

The National Science Foundation of China (31500308), the Zhejiang Provincial Natural Science Foundation of China (LQ16C040001) and the Zhejiang Science and Technology Innovation Program for College Students (2019R434006) funded this work. The funders had no role in study design, data collection and analysis, decision to publish, or preparation of the manuscript.

### Grant Disclosures

The following grant information was disclosed by the authors:
The National Science Foundation of China: 31500308.
Zhejiang Provincial Natural Science Foundation of China: LQ16C040001.
Zhejiang Science and Technology Innovation Program for College Students: 2019R434006.

### Competing Interests

The authors declare that they have no competing interests.

### Author Contributions

- Yun Tang conceived and designed the experiments, performed the experiments, analyzed the data, prepared figures and/or tables, authored or reviewed drafts of the paper, and approved the final draft.
- Zhi-Qiang Chen conceived and designed the experiments, performed the experiments, analyzed the data, authored or reviewed drafts of the paper, and approved the final draft.
- You-Fu Lin performed the experiments, analyzed the data, prepared figures and/or tables, authored or reviewed drafts of the paper, and approved the final draft.
- Jing-Yi Chen performed the experiments, authored or reviewed drafts of the paper, and approved the final draft.
- Guo-Hua Ding conceived and designed the experiments, analyzed the data, prepared figures and/or tables, authored or reviewed drafts of the paper, and approved the final draft.
- Xiang Ji conceived and designed the experiments, authored or reviewed drafts of the paper, and approved the final draft.

## Animal Ethics

The following information was supplied relating to ethical approvals (i.e., approving body and any reference numbers):

Our experimental procedures complied with the current laws on animal welfare and research in China, and were specifically approved by the Animal Research Ethics Committee of College of Ecology, Lishui University (Permit No. AREC-CELSU 201505).

## Data Availability

The raw data is available in the Supplemental Files.

## Supplemental Information

Supplemental information for this article can be found online at http://dx.doi.org/10.7717/peerj.8834#supplemental-information.

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
