# Peer review of "The combined effects of temperature and aromatase inhibitor on metamorphosis, growth, locomotion, and sex ratio of tiger frog (Hoplobatrachus rugulosus) tadpoles"

_PeerJ, doi:10.7717/peerj.8834_

## Round 0.1 · original submission · Major Revisions

I have now two reviews that suggest a number of flaws that should be solved in order to make this manuscript acceptable to be published in PeerJ. Both reviewers have doubts about the connection between the effects of the aromatase inhibitor, temperature and conservation strategies in the studied species. Please, explain why did you choose letrozole instead of the more commonly known aromatase inhibitors. Both reviewers have also pointed out a number of methodological problems and concerns about the validity of your findings that need your careful attention. I suggest taking all the reviewers' suggestions into full consideration.

·

Basic reporting

The paper is generally well-written and understandable. However, there are problems with tense and there are areas where the language is awkward making it difficult to read smoothly. For example the first line of the Introduction, line 46, there should be a modifier before ‘attention’, such as recent or much etc. and Line 89, ‘restrain the transformation’ ‘prevent’ would be a better word. The use of ‘E.g’ is problematic; the information would be better incorporated into a sentence.

Line 79: Whereas it is true that there has been quite a lot of work with steroidal hormones and amphibians it is ALSO true that there has been quite a bit of attention to the effects of aromatase inhibitors – contrary to what the authors state there is quite a bit of work with Xenopus and Silurana e.g. Olmstead et al. 2009 https://doi.org/10.1016/j.aquatox.2008.07.018, and Duarte-Gutterman et al. 2009 https://doi.org/10.1159/000280586

There appears to be an emphasis on Asian authors in the cited literature. Which is reasonable when referring specifically to the study organism, but may be limiting the breadth of the available information on the topic of aromatase inhibitor effects on phenotypic sex in general.

The authors should tell the reader why they chose letrozole instead of the more commonly known aromatase inhibitors (e.g. tamoxifen). It is particularly interesting that letrozole has a very specific mode of action (https://link.springer.com/article/10.1007/s10549-007-9696-3) and is more potent and specific for AI and is non steroidal.

Lines 91-95: Consistent with the title, the authors state clearly what the study entails. However, the last line of the Introduction leaves me wondering what the connection is between the two. I think this linkage of the effects of exposure to such a potent AI and conservation are a bit obscure and need explanation. Alternatively, remove the sentence.

Figure 1 has no SE; and shouldn’t it be + not just +? Legends for Figs 2 and 3 are switched. Would be helpful if there are no significant differences to state same. Are survival and % male means? Check language in legends.

Experimental design

There is no replication of the experimental units which is unfortunate and makes the results less rigorous. This should be addressed in the discussion.

Line 100 -104: The use of plastic in which to conduct the experiments is of concern to me. Recall that the topic of endocrine disrupting chemicals was in part ‘discovered’ by Ana Soto who observed her breast cells growing abnormally in plastic culture dishes.

Line 105: It is not clear how 1.8 tadpole density was maintained through metamorphosis…were mortalities replaced?

Line 111: “the feed..was only 100 mL …ethanol” needs to be reworded.

Line 118: how was the weighing of the tadpoles performed? Every X days they weighed 90 tadpoles in each treatment? Removed them to a tared beaker?

Line 123: It is not clear to me what happens with each tadpole that metamorphosed. Were the froglets put into individual lidded bowls or grouped? How many of the 90 froglets/tadpoles were tested for jumping; just 10? Just the first 10? It appears that your results are for 10 but there are data for 20 in your supplemental data. I’m confused. Please address this. Moreover, the supplemental data for the jumping results are only the means? Can you please provide the 3 distance measurements?

Line 128: is there a protocol that can be cited for the jumping test? If not, could it be explained in some more detail? It is not clear whether the froglets are jumping within the bowl or whether they are removed. Also, are they returned to a start position for the three jumps or are they probed to jump from where they landed from the previous jump?

Line 130: new paragraph at “The froglets were reared…”

Line 135: what is an appropriate amount of food? So, were froglets placed in pond after metamorphosis directly from jumping experiment? From exposure vessel? When did 90 days start? From beginning of exposure in plastic vessels; it says 90 days of feeding, I don’t understand how you could ensure 90 d for each froglet if they metamorphosed at different times? What happened at the end of the experiment? Were the ponds drained and the frogs recovered? How did they ensure all were recovered? Were the ponds enclosed to ensure that no frogs could escape?

Line 144: Define metamorphose rate? Did you move the tadpoles that had not metamorphosed into the pond when you moved others that had? I cannot follow the sequence of events.

How do the temperatures of 29c and 34c compare to the temperatures the frogs are naturally exposed to in their native habitat? Defend/explain why these were chosen.

Validity of the findings

The authors state that the exposure treatment did not affect metamorphosis rate nor survival rate. But I am not sure that this is accurate. I believe the authors have determined that there is no difference in metamorphosis rate nor survival rate AMONG the treatments. However, did they actually test to determine if the rates were different from 100% metamorphosis or survival? If they did the latter, then I can believe that there was no effect of temperatura or letrozole on survival or metamorphosis rate. (also see Line 188)

Could the authors please address the cause of the high mortality? The high mortality could have compromised the results. Is there any relationship between those tadpoles that did not metamorphose and those that died? What sex were the tadpoles that died? Could there have been a bias such that only or more females than males died?

Line 186: The authors indicate that their results show indirect support for sex hormones not involved in growth or development. This needs to be put into context within this early developmental window and at physiological concentrations. Outside of this window and at exposure to concentrations of estrogens or androgens greater than physiological concentrations there may in fact be effects on growth and development.

Line 195: The authors implicitly suggest they anticipated an AI to increase testosterone concentrations, which would increase the proportion of males to females and muscle development so that the frogs would jump further. On line 146 the authors considered SVL relative to jumping distance and did not observe a relationship. However, they did not consider mass or show the results of the relationship to jumping and mass. Lighter frogs may have been able to jump further, and there were effects of temperature on size.


Also, did the authors consider measuring testosterone and estradiol in the frogs?

Line 222: instead of accumulation, use additive. And instead of overlap use interact. Replication would have helped with defining this better.

Line 227: Statements such as ‘considered for the first time’ are unnecessarily grandiose when in the context of doing an experiment such as this where the AI effects on phenotypic sex of an amphibian generically have been explored and the only novelty is really the use of a different species I would strongly encourage the authors to leave this statement out so as to not turn off the reader.

Line: 232: “letrozole can induce a male bias in the tadpoles of H. rugulosus, and this effect is increased at high temperature” in Lines 170-175 I do not see that the proportion of males at letrozole and 34C are significantly higher than letrozole and 29C. Is this the case? Or is it that there was a sig difference in proportion between 29 and 34. And within temperatures sig diff between 0 letrozole and 0.1 and 1 treatments but not between temp x letrozole treatments? Do the authors know when the thermosensitive period is for this species?

Reviewer 2 ·

Basic reporting

.

Experimental design

.

Validity of the findings

.

Additional comments

“Effects of temperature and aromatase inhibitors on metamorphosis, growth, locomotion, and sex ratio in Hoplobatrachus rugulosus tadpoles”
General comments:
This is an original study focusing on the combined effect of temperature and an aromatase inhibitor over several aspects of growth, development, sex determination and performance of tadpoles and metamorphs of the anuran Hoplobatrachus rugulosus.

The study is well presented, however I prompt the authors to include more information linking why to study the effect of these two stressors (Temperature and aromatase inhibitor) in the context of this species (that is at same degree of risk). Also, considering the proposed framework they need to define more clearly the objectives of the study. For example, ending the Introduction I suggest authors to better present the objectives. Why to study the effect of aromatase inhibitor (letrozol) and temperature? How is the connection with conservation strategies in this species of high commercial value?
The experimental design is appropriate, it result clear and the analysis is well done.
Respect to the validity of the results, they are conclusive and clear.
Discussion in general is correct, but again, I suggest to focus and discuss after the objectives were clearly proposed.
Finally, be careful with grammar and punctuation.

Particular comments:
Introduction.
In the first paragraph, if authors are talking about amphibians in general, would be adequate to cite more general literature related to the effect of temperature and hormones on life history responses in amphibians.
Ending the Introduction, authors need to define the objectives of the study and link them with appropriate framework of study. Then, based on previous studies and the question/s authors propose here I suggest to better explain what is the purpose to use letrozol, Is it a common compound? Can it emulate responses in typical hormones? And how common is the use of this kid of substances in these studies? Then, it is not clear (both in the framework presented in Introduction and Discussion) why to study the effect of this aromatase inhibitor (letrozol) and temperature. As I suggested before, the connection between the effect of these stressors and the conservation status of this species need to be better explained.
In particular, experimental design: What was the criteria used for the set of experimental temperatures (29° and 34°)? These temperatures are common in nature? Or, It is related to settings at amphibian hatcheries? It must be better explained.
Discussion
Lines 186-188. This paragraph merely needs have connection with objectives. For example, authors suggest that letrozol results could be extrapolated to the effect of steroid and testosterone hormone as is observed in other studies, however, again I think it is needed to present the idea in the Introduction to connect with objectives and to be suitable discussed.
Lines 190-194. To discuss this result (discrepancy about the temperature range employed in this study) it is necessary to understand how the set of experimental temperature was determined.

Figures and legends.
There is a confusion between Fig. 2 and Fig. 3 and their legends. Please correct it.

---

## Round 0.2 · Minor Revisions

Thank you for your consideration of ours’ reviewers suggestions. I see yet some points that need your further attention.

In the subheading Results of the Abstract, you included some conclusions.

I would like you to include a consideration of the circumstances that could explain the contact between the frog species studied and letrozole.

I also think that you need a paragraph in the introduction section to explain why did you decide to study the influence of letrozole in locomotion. This could be easily accomplished by moving the paragraph in ln. 253 to 257 in the discussion section to the introduction.

In this version of your M&M you use a condition factor, which is crucial to interpret your results. You need to explain it more carefully, why did you decide to use it now?

Your results show no relationship between letrozole and jumping, thus your sentence in ln. 262: “However, the results of the present study are not enough to prove this conclusion..” concerning Nam et al. (2018) paper should be deleted.

I find that these paragraphs contradict each other:

> After controlling for the effect of the condition factor, neither temperature nor letrozole concentration affected jumping ability, suggesting that temperature and steroid hormone do not significantly affect the locomotion (Discussion)


> (1) high temperature can accelerate the growth and development of H. rugulosus tadpoles, shorten the metamorphosis time, strengthen jumping ability, and increase the proportion of males… (Conclusions)


Did or did not the temperature influence jumping abilities? Please, be more careful in writing to avoid misunderstandings.

·

Basic reporting

revision is acceptable

Experimental design

revision is acceptable

Validity of the findings

revision is acceptable

Additional comments

revision is acceptable

---

## Round 0.3 · Major Revisions

I am sorry for the delay, but I had still doubts about some critical points that required further attention. I decided to send the manuscript back to Reviewer 2 to look at again, and I agree with our reviewer. Please take these new comments in full consideration.

Reviewer 2 ·

Basic reporting

This is an original study focusing on the combined effect of temperature and an aromatase inhibitor (letrozole) on several aspects of growth, development, sex determination and performance of tadpoles and metamorphs of the anuran Hoplobatrachus rugulosus.

My main concerns are focused on the objectives and how the discussion was addressed.

Experimental design

My main concern is on the scientific questions of the study. It is not clear why the use of litrozole, Is it a common substance found in nature? or Is it commonly used in the industry of frog-farms? Then, I could not find the link with temperature and the interest of the study.

Validity of the findings

My suggestion is that authors need to explain how your results relate to expectations (Was letrozol a disruptor of life history patterns? Is there an interaction with temperature? Can you explain the importance of the results on the conservation status of the species? And for the economic purpose?), to the literature (including more cites) and how they are important in the field of conservation of the species.

Additional comments

Review
“Effects of temperature and aromatase inhibitors on metamorphosis, growth, locomotion, and sex ratio in Hoplobatrachus rugulosus tadpoles”

General comments:
This is an original study focusing on the combined effect of temperature and an aromatase inhibitor (letrozole) on several aspects of growth, development, sex determination and performance of tadpoles and metamorphs of the anuran Hoplobatrachus rugulosus.
My main concern is on the scientific questions of the study. It is not clear why the use of litrozole, Is it a common substance found in nature? or Is it commonly used in the industry of frog-farms? Then, I could not find the link with temperature and the interest of the study.

Then, my main concerns are focused on the objectives and how the discussion was addressed.

Particular comments

Line23-25: Please correct the text to:
In the present study, H. rugulosus tadpoles were subjected to two water temperatures (29 °C and 34 °C) and three letrozole concentrations in feed (0 mg/g, 0.1 mg/g, and 1 mg/g) to examine the effects of …

Line 31. Please, Could you specify the difference between body size and condition? I do not find a diference.

Line 34. Again, what is a condition factor? There some words and statements (“condition,” “condition values” and “condition factor”) that need to be explained. In the section Data measurement “condition factor” is well explained, then I suggest to include here a kind of similar explanation to put the reader in context.

Introduction
Lines 45-53 I suggest that authors need to connect the use of letrozole in the questions that they are proposing. Is letrozole common in nature? It is a problem to this protected species?

Lines 53-54. Please include citations.

Line 66, 105. Please correct, the cite is. Hayes, 1997

Lines 119-124. I suggest that authors need to connect the use of letrozol in the questions that they are proposing. Is letrozole common in nature? It is a problem to this protected species?
Lines 250-252. I think this conclusion must be taken with caution.


Discussion
In the first paragraph authors present the relationship between temperature and life history responses, and I found that literature cited is insuficient. I suggest to include more general literature as e.g Stearns work (Stearns, S. C. 1992. The evolution of life histories.
Stearns, S. C. and J. C. Koella. 1986. Evolution and some others). There is a lot of literature demonstrating that temperature increase development and result in smaller froglets. Then, I suggest authors to consider include more references.

In general Discussion needs to be reorder. Organize the Discussion from the specific to the general.
In the Introduction authors pay attention on the economic value of this anuran species having in account the context of their conservation status.
I think the main problem with this manuscript is that objectives are not clear enough and there is no relation between the results and there is no a clear what are the implications of the findings.

Authors need to clarify what is the objective to use letrozole and why temperature is also important in the interaction with this aromatase inhibitor. Is letrozole common in nature? The use in the study has the purpose to mimic other aromatase inhibitors? Is it commonly used in frog-farms?
Once the objectives are clear try to connect the results in this sense to answer the questions posed in the Introduction. I recommend to explain how the results support the objectives (once they are clarified) and answers proposed and, how these answers fit in with existing knowledge on the topic. By the way, because of the abundant published work related to this topic I suggest to include more general cites (not just recent and regional research) to discuss the results.
Also, authors need to explain how your results relate to expectations (Was letrozol a disruptor of life history patterns? Is there an interaction with temperature? Can you explain the importance of the results on the conservation status of the species? And for the economic purpose?), to the literature (including more cites) and how they are important in the field of conservation of the species.

---

## Round 0.4 · Major Revisions

In my view, most of the reviewers' concerns have not been satisfactorily addressed. One of the main questions remains. By which I mean specifically, you need to explain under what circumstances can H. rugulosus contact letrozole or any other aromatase inhibitor. Are these substances products of water pollution in some of the environments where this species lives? Is it a common substance found in nature? Or Is it commonly used in the industry of frog-farms? You have stressed that aromatase inhibitors have become more widely used, without explaining what this use is.
About the condition factor, you stressed that it "was defined as the body mass divided by the SVL" by whom? You need at least one quotation here.

You did not make the Stern quotations, as was suggested by our reviewer.

Please insert space before brackets in lines 81 and 91.
Please, in line 92 of the pdf, it is stressed: "Aromatase inhibitors comprise a class of synthetic drugs that can inhibit the activity of aromatase (Li et al., 2007)" This sentence should be rephrased as an aromatase inhibitor inhibits the activity of the aromatase by definition.
In the discussion section, you stressed: "However, we only wanted to confirm whether different temperature treatments are related to the growth and development of H. rugulosus, more experiments are thus
needed to further explore the effect of different temperature treatments on amphibian locomotion ability." This paragraph sounds as if you never intended to take locomotion into account. However, you do include this issue in M&M and Results sections. Please, rephrase this paragraph.
The following paragraph is not a conclusion: "The effects of temperature and aromatase inhibitors on phenotypic plasticity of H. rugulosus tadpoles, especially on gender phenotype, were considered in this experiment. Besides, the effects of aromatase inhibitors on the growth, development and locomotion of H. rugulosus tadpoles were examined." Please begin this section in "(1) high temperature can accelerate...
I have read your paper many times. It is tough to follow the logic of your research, and this point must be improved.

I am concerned about your paper because I feel that we are moving in circles. Your study needs fluency, and our reviewers have contributed significantly to help you. I strongly encourage you to take the last observations into account, or I will be unable to accept your work.

---

## Round 0.5 · accepted · Accept

Please, the next sentence should be rewritten as follow:

Aromatase inhibitors comprise a class of synthetic drugs that, apart from inhibit the activity of aromatase (Li et al., 2007), block the transformation of testosterone to estradiol, and reverse the sex transition from female to male or masculinize the gonads (Yu et al., 1993; Chardard & Dournon, 1999; Miyata & Kubo, 2000; Yu et al., 1993)